# Role-Engineering Optimization with Cardinality Constraints and User-Oriented Mutually Exclusive Constraints

**Wei Sun *** , **Hui Su and Hongbing Liu**

Center of Network Information and Computing, Xinyang Normal University, Xinyang 464000, China;
suhuixy@xynu.edu.cn (H.S.); liuhbing@xynu.edu.cn (H.L.)
*   Correspondence: sunny810715@xynu.edu.cn

**Abstract:** Role-based access control (RBAC) is one of the most popular access-control mechanisms because of its convenience for management and various security policies, such as cardinality constraints, mutually exclusive constraints, and user-capability constraints. Role-engineering technology is an effective method to construct RBAC systems. However, mining scales are very large, and there are redundancies in the mining results. Furthermore, conventional role-engineering methods not only do not consider more than one cardinality constraint, but also cannot ensure authorization security. To address these issues, this paper proposes a novel method called role-engineering optimization with cardinality constraints and user-oriented mutually exclusive constraints (REO_CCUMEC). First, we convert the basic role mining into a clustering problem, based on the similarities between users and use-partitioning and compression technologies, in order to eliminate redundancies, while maintaining its usability for mining roles. Second, we present three role-optimization problems and the corresponding algorithms for satisfying single or double cardinality constraints. Third, in order to evaluate the performance of authorizations in a role-engineering system, the maximal role assignments are implemented, while satisfying multiple security constraints. The theoretical analyses and experiments demonstrate the accuracy, effectiveness, and efficiency of the proposed method.

**Keywords:** role engineering; role mining; role assignments; cardinality constraints; user-oriented mutually exclusive constraints

---

## 1. Introduction

With the rapid development and comprehensive application of network-information technology, there are considerable amounts of storage and many exchanges in large-scale and complex information-management systems [1]. Determining how to ensure the security of system data and user information has attracted much interest. Numerous enterprises and organizations have adopted role-based access control (RBAC) as their main access-control mechanism, since the employment of RBAC is not only convenient and flexible, but also reduces the computational complexity of problems and alleviates the management burdens of systems [2–7]. With the successful implementation of RBAC systems, devising an accurate and effective set of roles and constructing a good RBAC system, that can satisfy actual application requirements, have become critical tasks. Role-engineering technology [8,9], which aims to migrate from non-RBAC systems to RBAC systems, has been proposed. There are two main approaches to this process: Top-down [10] and bottom-up [11–15]. The former devises roles by analyzing and decomposing business processes into smaller units that are associated with the needed permissions. However, this approach is time-consuming and labor-intensive when there are tens of



thousands of users and millions of permissions. The latter starts from the original user-permission assignments and aggregates them into roles by applying data mining techniques, which is also known as role mining. This latter approach has gained considerable attention in recent years.

To discover interesting roles from existing permission assignments, two algorithms called the Complete Miner and Fast Miner are proposed [16]. Both algorithms use subset enumeration and allow overlapping roles. While the first algorithm can enumerate all potential roles, its computational complexity is exponential. The second algorithm improves the mining process, and its computational complexity is remarkably reduced. However, it identifies only a partial set of roles. The Fast Miner is sufficient for practical applications. Vaidya et al. [17] converted role mining into a matrix-decomposition problem and presented a definition for a basic role mining problem (basic RMP). Basic RMP has been proven to be *NP*-complete, for which several existing studies have already been done to find efficient solutions. According to different optimization objectives for role mining, many other approaches have been proposed, such as $\delta$-approx RMP [17], min-noise RMP [17], edge RMP [18], usage RMP [19], and user-oriented exact RMP [20].

Essentially, role mining is the task of clustering users with identical or similar permissions and constructing different roles with these permissions. Indeed, many roles contain several identical permissions and are frequently assigned to users. Frequently usable roles can facilitate the management and maintenance of the system and decompose the set of users into clusters of users with different attribute characteristics [21]. However, the analysis of mining, resulting from large-scale clusters, is complex [22]. On the other hand, a fairly small number of roles may not be assigned or assigned to only a small number of users. These roles are not frequently used, so they are redundant roles. Thus, owing to the diversity of system resources and the variability of resource access, there are redundancies in the mining results that use conventional methods.

A key characteristic of RBAC is that it allows the specification and enforcement of various security policies [23–25], such as cardinality constraints, which can reflect the security policies of different organizations and ensure system security. For example, the general-manager role in a company must be assigned to only one person; ordinary users should not have too many roles, otherwise there is the possibility for users to abuse their privileges (e.g., the fewer roles assigned to the permission of opening a safe, the better). There are four different types of cardinality constraints [26]: (1) User-role cardinality constraint (UCC), (2) permission-role cardinality constraint (PCC), (3) role-user cardinality constraint (RUC), and (4) role-permission cardinality constraint (RPC). In the approaches for role optimization with cardinality constraints, most existing methods not only do not consider more than one constraint, but also cannot determine whether other security constraints are met in the constructed RBAC system.

Furthermore, mutually exclusive constraints and user-capability constraints are also essential components in the enforcement of security policies, especially in the process of role assignments. The most commonly used constraint, which is called the statically-mutually-exclusive-roles (SMER) constraint, aims at restricting the role memberships assigned to a single user [27]. For example, the account-manager and financial-auditor roles cannot be assigned to the same person. Another important constraint, which is usually used in actual application environments, is the user-capability constraint [28]. Specifically, users cannot be assigned to roles arbitrarily in an organization, since different users have different qualifications and competencies. For example, a user with a degree in business can perform roles, such as an account manager or financial auditor, but cannot perform the roles of a computer professional, such as the role of a software designer, software developer, or software tester. The premise of implementing role assignments is that an RBAC system already exists. However, in many cases, the systems and constraints are completely unknown.

To address the above issues, this paper proposes a novel method called role-engineering optimization with cardinality constraints and user-oriented mutually exclusive constraints (REO_CCUMEC). The main contributions of this paper are as follows:

(1)　Partitioning and compression are two important methods used to analyze clustering problems; they are widely used in scientific research and production practice because of their simple

and accurate characteristics [29]. In order to reduce computational complexity and mining scale, we convert the basic role mining problem into a clustering problem, use partitioning and compressing technologies to eliminate redundancies, and evaluate the accuracy of the proposed method.

(2)　Role optimization that satisfies one cardinality constraint may violate another cardinality constraint. In order to limit the number of roles assigned to any user and/or a related permission, we present three role-optimization problems and their corresponding algorithms, and evaluate the effectiveness of the proposed method.

(3)　Mutually exclusive constraints, user-capability constraints, and cardinality constraints are critical to ensure authorization security. In order to satisfy these constraints, while maximizing the role assignments in the role-engineering system, we present a role-assignment algorithm and evaluate the efficiency of the proposed method.

The remainder of the paper is organized as follows. In Section 2, we discuss the related work and present preliminary information. Section 3 proposes a novel method that includes three aspects: Pre-processing, role optimization, and role assignments. We present theoretical analyses and examples in Section 4. We show the experimental evaluations in Section 5. Section 6 concludes the paper and discusses future work.

## 2. Related Work and Preliminary Information

### 2.1. Methods of Role Optimization

Many methods have been proposed for role optimization. Depending on whether or not constraints are considered in role optimization, existing studies mainly fall into the following two categories: Role optimization with no constraints and role optimization with constraints.

Vaidya et al. converted role mining into a matrix-decomposition problem and presented a definition for a basic role mining problem, that attempts to find a minimal set of roles from bottom-user permission assignments and completely cover the original assignments. However, it is difficult to derive an optimal role set in practical applications. To reflect the organization-function requirements and enhance the interpretability of mining roles, Molloy et al. [30] represented roles with the formal concept of lattices and proposed an optimization algorithm. To optimize the RBAC system, Zhang et al. [31] presented an algorithm using graph-optimization theory. However, this algorithm did not eliminate the redundancies in the mining results. Ene et al. [32] adopted heuristics and graph theory to mine as few optimized roles as possible, thereby reducing the redundancies of the mining roles. Lu et al. [19] proposed a unified role-optimization framework and presented a number of greedy algorithms that could solve basic RMP, δ-approx RMP, min-noise RMP, and edge RMP, based on methods of integer linear programming and Boolean matrix decomposition. However, the scale of role optimization is very large. To reduce the complexity of solving problems, Colantonio et al. [33] divided the user-permission-assignment dataset into several subsets. To reduce the mining scale, Verde et al. [34] converted role mining into a clustering problem, which compresses the division into a single sample, extracts similar features from multiple divisions, and ensures the integrity of the mining results. Although, constraints are essential for the RBAC model, none of these methods take constraints into consideration.

In order to avoid an abuse of privileges, Kumar et al. [35] proposed a constrained role-miner algorithm, that limits the number of permissions assigned to a role. Blundo et al. [36] proposed a heuristic capable of returning a complete set of roles, thereby satisfying the same cardinality constraint as above. Hingankar et al. [37] proposed a biclique-cover method to derive roles that limit the maximum number of users related to a role. John et al. proposed two alternative approaches for restricting the number of roles assigned to a user: Role priority-based approach (RPA) and the coverage of permissions-based approach (CPA). The RPA prioritizes roles based on the number of permissions and assigns optimal roles to users, according to the priority order. The CPA chooses roles by iteratively

picking the role with the largest number of permissions that are yet uncovered and then ensures that no user is assigned more than a given number of roles [38]. In order to limit the maximum number of users or permissions related to a role, Ma et al. [26] proposed a role mining algorithm to generate roles based on permission cardinality constraints and user cardinality constraints. In order to simultaneously limit the maximum number of roles assigned to a user and a related permission, Harika et al. proposed the two role-optimization methods: Post processing and concurrent processing. In the first method, roles are initially mined without taking the constraints into account. The user-role and role-permission assignments are then checked for constraint violation in the optimization process and appropriately re-assigned, if necessary [39]. The concurrent processing method implements optimization with double constraints during the process of role mining. In addition to these methods for satisfying cardinality constraints, Sarana et al. [40] proposed three role-optimization methods, including separation-of-duty constraints either, during, or after, the mining process. In order to satisfy separation-of-duty constraints and ensure authorization security, Sun et al. [41] proposed a method called role-mining optimization, with separation-of-duty constraints and security detection for authorizations.

### 2.2. Methods of Role Assignments

In order to obtain permissions, while satisfying a collection of constraints for a given authorization request, Zhang et al. proposed a user authorization query (UAQ) problem, that adopts the greedy algorithm and mutually-exclusive-role constraint to search for objects. The UAQ is used to discover a set of roles to be activated in a single session for the particular set of permissions requested by the user [42]. Lu et al. [43] proposed a novel approach, based on role–permission reassignments, to support the UAQ, which assists administrators in modifying system configurations in an automatic manner. In order to implement role assignments, while satisfying the *t-t* SMER constraints in RBAC, Roy et al. proposed a method for finding the minimum number of users with multiple *t-t* SMER constraints, modelled the general problem using graphs, and presented a two-step method for solving the problem [44]. Afterwards, the problem of the cardinality constrained-mutually exclusive task for minimum users, was defined. This problem aims to find the minimum users that can carry out a set of tasks, while satisfying the given security constraints [45]. Furthermore, Roy et al. [28] defined the employee assignment problem, which aims to assign employees to roles, so that the maximal flexibility is reflected in assigning roles to employees, while ensuring that the user-capability constraints, role-cardinality constraints, and liveness constraints are met simultaneously.

Obviously, from the above analyses, we find that there are three limitations in the existing studies. The first limitation is that the role-mining scale is very large, and there are redundancies in the mining results. The second limitation is that most existing role-optimization methods only consider one cardinality constraint and do not evaluate authorization security in a constructed RBAC system, so role assignments cannot satisfy user-capability constraints and mutually exclusive constraints. The third limitation is that the existing role assignments assume that RBAC systems already exist. However, in many cases, the systems are completely unknown, and the constraints are uncertain. Hence, in this paper, we propose a novel role-engineering method (REO_CCUMEC), which mainly includes three elements: (1) Partitioning and compressing technologies are used to eliminate redundancies for the unconstrained role mining, (2) the role-optimization problems and their corresponding algorithms are presented for satisfying double cardinality constraints simultaneously, and (3) the maximal role assignments are implemented, while satisfying multiple security constraints in the constructed RBAC system. We also evaluate the performance of the proposed method using three groups of experiments and present its advantages and limitations.

### 2.3. Preliminaries

#### 2.3.1. Basic Components of Role Engineering

Conventional role engineering consists of the following basic components [5–7]:

(1)   $U$, $P$, and $R$ are the basic elements of RBAC; these elements denote a set of users, a set of permissions, and a set of roles, respectively;

(2)   $UPA \subseteq U \times P$, a many-to-many mapping of user-permission assignments in the non-RBAC model;

(3)   $UA \subseteq U \times R$, a many-to-many mapping of user-role assignments in the RBAC model;

(4)   $PA \subseteq R \times P$, a many-to-many mapping of role-permission assignments in the RBAC model;

(5)   $user\_roles(u) = \{r | \exists r \in R : (u, r) \in UA\}$, the mapping of user $u$ onto a set of roles;

(6)   $role\_users(r) = \{u | \exists u \in U : (u, r) \in UA\}$, the mapping of role $r$ onto a set of users;

(7)   $role\_permissions(r) = \{p | \exists p \in P : (r, p) \in PA\}$, the mapping of role $r$ onto a set of permissions;

(8)   $permission\_roles(p) = \{r | \exists r \in R : (r, p) \in PA\}$, the mapping of permission $p$ onto a set of roles;

(9)   $user\_permissions(u) = \{p | \exists p \in P, \exists r \in R : ((u, r) \in UA) \wedge ((r, p) \in PA)\}$, the mapping of user $u$ onto a set of permissions.

### 2.3.2. RBAC Constraints

We consider different kinds of constraints in RBAC: UCC and PCC, mutually exclusive constraints, and user-capability constraints.

(1) The UCC and PCC

The UCC [26] states that, for a given set $U$ of users, set $R$ of roles, and threshold $MRC_{user}$, the number of roles assigned to any user should not exceed $MRC_{user}$. This can be formalized as follows:

$$\forall u \in U : \left| user\_roles(u) \cap R \right| \leq MRC_{user}. \tag{1}$$

The PCC [26] states that, for a given set of $U$ of users, set $R$ of roles, and threshold $MRC_{permission}$, the number of roles to which any permission can be assigned should not exceed $MRC_{permission}$. This can be formalized as follows:

$$\forall p \in P : \left| permission\_roles(p) \cap R \right| \leq MRC_{permission}. \tag{2}$$

In addition, as the mapping relationships of $UA$ and $PA$ are bidirectional, there are another two constraints that, respectively restrict the number of users and the number of permissions assigned to a role, which are not discussed in the paper.

(2) Mutually exclusive constraints

According to the different intensities of restrictions, the SMER constraint includes the following two types [27]:

The *t-m* SMER constraint states that, given $m$ roles $r_1, r_2, \ldots, r_m$, no user is allowed to have $t$ or more of these $m$ roles. This constrained is expressed as $smer < \{r_1, r_2, \ldots, r_m\}, t >$, where $m$ and $t$ are integers that satisfy $2 \leq t \leq m$. This can be formalized as follows:

$$\forall u \in U : \left| \{r_1, r_2, \ldots, r_m\} \cap user\_roles(u) \right| < t. \tag{3}$$

The *t-t* SMER constraint states that, given $t$ roles $r_1, r_2, \ldots, r_t$, no user is allowed to have all of these $t$ roles. This is expressed as $smer < \{r_1, r_2, \ldots, r_t\}, t >$, where $t$ is an integer, and $t \geq 2$, and can be formalized as follows:

$$\forall u \in U : \{r_1, r_2, \ldots, r_t\} user\_roles(u). \tag{4}$$

It has been shown that any *t-m* SMER constraint can be equivalently represented as a set of *t-t* SMER constraints [27]. Thus, we only take into consideration the *t-t* SMER constraint in this paper.

(3) User-capability constraint

This constraint is represented as the Boolean matrix *UC* [28], where the rows correspond to users, and the columns correspond to roles. The value 1 in cell *UC*[*i*][*j*] denotes that user $u_i$ is capable of performing role $r_j$; otherwise, $u_i$ cannot perform $r_j$. This can be formalized as follows:

$$UC[i][j] = \begin{cases} 1, if\ u_i\ is\ capable\ of\ performing\ r_j \\ 0, otherwise \end{cases} \tag{5}$$

### 2.3.3. Similarity and Dissimilarity in Clustering

The Jaccard coefficient [46] in statistic, which is widely used to measure the similarity or dissimilarity (also called distance) among different sets of samples, aims to identify sample clusters. Given set $S = \{S_a, S_b, \dots, S_i, \dots\}$, where $S_a = \{a_1, a_2, \dots\}$, $S_b = \{b_1, b_2, \dots\}$, $S_i = \{i_1, i_2, \dots\}$,

(1) $\forall(S_i, S_j) \in S$; the similarity and dissimilarity between sample $S_i$ and sample $S_j$ are, respectively, calculated as follows:

$$sim(S_i, S_j) = \frac{|S_i \cap S_j|}{|S_i \cup S_j|} \tag{6}$$

$$dis(S_i, S_j) = 1 - sim(S_i, S_j) \tag{7}$$

(2) $\forall(S_i, S_{j1}, S_{j2}, \dots) \in S$; the similarity and dissimilarity between sample $S_i$ and sample set $\{S_{j1}, S_{j2}, \dots\}$ are, respectively, calculated as follows:

$$sim(S_i, \{S_{j1}, S_{j2}, \dots\}) = \frac{1}{|\{S_{j1}, S_{j2}, \dots\}|} \sum_{S_j \in \{S_{j1}, S_{j2}, \dots\}} sim(S_i, S_j) \tag{8}$$

$$dis(S_i, \{S_{j1}, S_{j2}, \dots\}) = 1 - sim(S_i, \{S_{j1}, S_{j2}, \dots\}). \tag{9}$$

### 2.3.4. Basic RMP Problem and the Fast Miner Method

The basic RMP [17] can be formalized as follows:

$$\begin{cases} \min|R| \\ UA \otimes PA = UPA \end{cases} \tag{10}$$

For the sake of simplicity, the *UPA*, *UA*, and *PA* are used to represent their respective assignment relationships, as well as the corresponding matrices. The Fast Miner method [16] mainly consists of the following two steps:

**Step 1.** Based on the hash mapping rule, a group of all users who have the exact same set of permissions for a given permission assignment, and construct an initial set of roles. This significantly reduces the size of the original data set.

**Step 2.** Identify all potentially interesting roles by implementing intersections between all pairs of the initial roles. Generate new roles and count the number of users associated with any new role.

For readability, we have summarized the main symbols used in the paper in Table 1.

**Table 1.** Main symbols and their meanings.

| Symbol | Meaning |
|---|---|
| *U,P,R,UA,PA,UPA* | Basic components of RBAC |
| UCC | Limitation on the number of roles assigned to any user |
| $MRC_{user}$ | Threshold of the UCC |
| PCC | Limitation the number of roles related to any permission |
| $MRC_{permission}$ | Threshold of the PCC |
| SMER | Static mutually exclusive roles |
| $smer<\{r_1,r_2,\dots,r_t\},t>$ | *t-t* SMER constraint |
| C | Set of the *t-t* SMER constraints |
| UC | Matrix of the user-capability constraints |
| CU | Cluster of users |
| VC | Set of the compression points |
| RR | Role-utilization ratio |

## 3. Proposed Method

In this section, we propose a novel method, REO_CCUMEC, which includes three elements: (1) Preprocessing for basic RMP, (2) role optimization satisfying cardinality constraints, and (3) role assignments satisfying multiple constraints.

### 3.1. Preprocessing for Basic RMP

To satisfy the basic RMP, the methods of the Fast Miner algorithm and Boolean matrix decomposition are used to mine the initial roles, as shown in Algorithm 1.

---
**Algorithm 1.** Initial role mining for basic RMP.

---
**Input:** the original matrix *UPA*
**Output:** preprocessed matrices *UA* and *PA* and the initial set *CR* of the roles
The Fast Miner and Boolean matrix decomposition are adopted to derive *CR* and configure RBAC, such that
$$\begin{cases} \min|CR| \\ UA \otimes PA = UPA \end{cases}.$$

---

According to Equation (6) and the results from Algorithm 1, the similarity and dissimilarity between $u_i$ and $u_j$ are calculated as follows:

$$sim(u_i, u_j) = \frac{\left|user\_permissions(u_i) \cap user\_permissions(u_j)\right|}{\left|user\_permissions(u_i) \cup user\_permissions(u_j)\right|} \tag{11}$$

$$dis(u_i, u_j) = 1 - sim(u_i, u_j). \tag{12}$$

Partitioning can be done in many ways. However, it has been shown that, using business information is typically preferable to using other types of information, since it generates more meaningful roles. Business information includes both user and permission attributes. For the sake of clarity, we only consider partitions induced by user attributes.

According to the above calculations, we can identify the cluster {$CU_1,CU_2,...$} of users. Then, we use the partitioning and compressing technologies to handle each user cluster independently.

#### 3.1.1. Partitioning User Clusters

To identify user clusters, we use the well-known clustering algorithm, partitioning around cluster medoids (PAM) [47]. This algorithm is similar to the *k*-means clustering algorithm, except that dissimilarities are used instead of distances, and center points are used instead of means. First, we define the center point as follows.

**Definition 1.** *(center point) Given a cluster of users $CU = \{u_1, u_2, \dots, u_i, \dots, u_j, \dots\}$, the user $u_i$ is called the center point if and only if:*

$$\forall u_j \in CU \setminus \{u_i\} : dis(u_i, CU \setminus \{u_i\}) < dis(u_j, CU \setminus \{u_j\}). \tag{13}$$

In other words, the center point of a cluster is the user whose dissimilarity to all the other users in the cluster is minimal. The partitioning process is presented in Algorithm 2.

---

**Algorithm 2.** Partitioning user clusters.

---

**Input:** user cluster *CU* and the *k* number of center points
**Output:** center points and partitions
1. Randomly choose *k* users $u_1, u_2, \dots, u_k$ in *CU* as the initial center points;
2.   **for** each center point $u_i$ in $\{u_1, u_2, \dots, u_k\}$ **do**
3.     **for** each non-center point $u_j$ in $associate(u_i)$ **do**
4.       $dis(u_i, associate(u_i)) = 1 - \frac{1}{|associate(u_i)|} \sum\limits_{u_j \in associate(u_i)} sim(u_i, u_j);$
5.       $dis(u_j, associate(u_i) \setminus \{u_j\} \cup \{u_i\}) = 1 - \frac{1}{|associate(u_i) \setminus \{u_j\} \cup \{u_i\}|} \sum\limits_{u_k \in (associate(u_i) \setminus \{u_j\} \cup \{u_i\})} sim(u_j, u_k);$
6.       **if** $dis(u_j, associate(u_i) \setminus \{u_j\} \cup \{u_i\}) < dis(u_i, associate(u_i))$ **then**
7.         $associate(u_i) = associate(u_i) \setminus \{u_j\} \cup \{u_i\};$
8.         $swap(u_j, u_i)$ and divide *CU* into *k* partitions;
9.       **end if**
10.   **end for**
11. **end for**

---

In Algorithm 2, we first randomly select *k* users $u_1, u_2, \dots, u_k$, take them as the initial center points, and divide *CU* into *k* partitions (line 1). The function $associate(u_i)$ represents all the non-center points closest to center point $u_i$ in a partition, and we associate each user to the closest center point. Then, for each center point $u_i$ and non-center point $u_j$ in $associate(u_i)$, we respectively calculate $dis(u_i, associate(u_i))$, $dis(u_j, associate(u_i) \setminus \{u_j\} \cup \{u_i\})$ in lines 2–5. Lines 6–9 indicate that if the dissimilarity of $u_j$ to the set of $associate(u_i) \setminus \{u_j\} \cup \{u_i\}$ is less than that of $u_i$ after swapping the two users, then $u_j$ is referred to as the new center point instead of $u_i$.

Computational complexity: Partitioning a user cluster depends on the double loops and the swap operations. The execution time of the algorithm is $O(s \times k \times (n - k))$, where k is the number of center points, n is the number of users in the cluster, and s is the number of swaps between the center points and non-center points. Usually, $k \ll n$, so the impact of k on the performance can be ignored, and the total time is really $O(s \times n)$.

3.1.2. Compressing Cluster Partitions

After identifying user clusters and the respective center points, we further simplify each cluster using the support degree and compression point, which are defined as follows.

**Definition 2.** *(support degree of a permission) Let the user cluster be $CU = \{u_1, u_2, \dots, u_i, \dots\}$, where $user\_permissions(u_i) = \{p_{i1}, p_{i2}, \dots, p_{ij}, \dots\}$, and $p_{ij}$ is the permission possessed by $u_i$. The percentage of different users possessing p in CU is called the support degree of p with respect to CU. This percentage is represented as,*

$$support_{CU}(p) = \frac{|\{u_k | \exists u_k \in CU : p \in user\_permissions(u_k)\}|}{|CU|} \tag{14}$$

*where $support_{CU}(p) \in (0, 1]$.*

**Definition 3.** *(compression point) Given user cluster CU = {u₁,u₂, … ,uᵢ, … } and threshold t, user uᵢ is called the compression point if and only if*

$$\exists u_i \in CU, \forall p_{ij} \in u_i : support_{CU}(p_{ij}) \geq t. \tag{15}$$

We represent the cluster with $u_i$, and the sets of all compression points are represented as *VC*. The compression process is presented in Algorithm 3.

---

**Algorithm 3.** Compression cluster partitions.

---

**Input:** the initial set *CR* of the roles, the set *VC* of the compression points, the partition with center point $u_i$, and threshold *t*

**Output:** compressed matrix $UPA_{compressed}$

1. Initialize $UPA_{compressed} = \Phi$, $VC = \Phi$;
2.   **for** each $p$ in *CR* **do**
3.       $support_{associate(u_i) \cup \{u_i\}}(p) = \frac{|\{u|\exists u \in associate(u_i) \cup \{u_i\}: p \in user\_permissions(u)\}|}{|associate(u_i) \cup \{u_i\}|}$;
4.     **if** $support_{associate(u_i) \cup \{u_i\}}(p) \geq t$ **then**
5.       insert $u_i$ into *VC*;
6.       $UPA_{compressed} = UPA_{compressed} \cup \{(u_i, p)\}$;
7.     **end if**
8.   **end for**

---

Indeed, threshold *t* plays an important role in identifying the compression point. For example, when *t* equals 1, it is difficult to identify the compression point, because of the differences among users; when *t* is less than 1, identification is possibly easier.

*3.2. Role Optimization Satisfying Cardinality Constraints*

In order to optimize the pre-processed results, the UCC and PCC should be taken into consideration individually or simultaneously in role optimization. Specifically, the preprocessed matrices *UA* and/or *PA* are first checked to determine if they violate the given cardinality constraint(s). If there are no constraint violations, they are regarded as efficient solutions. Otherwise, to limit the number of roles assigned to the violating user and/or the related permission, we next present three role-optimization problems and the corresponding algorithms.

3.2.1. Role Optimization Satisfying UCC

**Definition 4.** *(a role-optimization problem with UCC) Given a user-permission assignment matrix UPA_{n×m}, the preprocessed UA and PA matrices, and a particular threshold MRC_{user}, find an optimal set R of roles, such that the UA and PA are consistent with the UPA, the number of roles assigned to any user is less than or equal to MRC_{user}, and the number of the optimal roles is minimized. This process can be formalized as follows:*

$$\begin{cases} \min|R| \\ UA \otimes PA = UPA \\ \sum_j UA[i][j] \leq MRC_{user} \leq |R|, \forall i \in [1, n] \end{cases}. \tag{16}$$

According to Definition 4, in order to satisfy role optimization in the presence of UCC, the optimizing process is presented in Algorithm 4.

---

**Algorithm 4.** Role optimization satisfying UCC.

---

**Input:** preprocessed matrices *UA* and *PA*, the initial role set *CR*, and threshold $MRC_{user}$
**Output:** the optimized matrices *UA* and *PA*
1. Define and compute *count_user_roles(u)* as the number of roles possessed by user *u*;
2. Define and compute *count_role_users(r)* as the number of users assigned to role *r*;
3. **while** $\exists u \in U$: *count_user_roles(u)* > $MRC_{user}$ **do**
4.    $k$ = *count_user_roles(u)* − ($MRC_{user}$ − 1);
5.    Choose the top *k* roles from *u* with the highest *count_role_users(r)* values to constitute set *S*;
6.    Merge the permissions of all the *k* roles and denote the union as set $P_S$;
7.    Create a new role $r_{nr}$ such that *role_permissions($r_{nr}$)* = $P_S$;
8.    **for** each $p_t$ in *P* **do**
9.     **if** $p_t \in P_S$ **then**
10.      *PA[nr][t]* = 1;
11.     **else**
12.      *PA[nr][t]* = 0;
13.     **end if**
14.    **end for**
15.    **for** each $u_i$ in *U* **do**
16.     **if** $\forall r_j \in S$: *UA[i][j]* = = 1 **then**
17.      $\forall r_j \in S$: *UA[i][j]* = 0;
18.      *UA[i][nr]* = 1;
19.     **else**
20.      *UA[i][nr]* = 0;
21.     **end if**
22.    **end for**
23.    Update *count_user_roles(u)* and *count_role_users(r)*;
24. **end while**

---

In Algorithm 4, we first define two functions, *count_user_roles(u)* and *count_role_users(r)* (lines 1–2). Line 3 determines whether the number of roles possessed by any user exceeds $MRC_{user}$. For each violating user *u*, the *k* number of roles is represented by calculating *count_user_roles(u)* − ($MRC_{user}$ − 1) in line 4, and the top *k* roles, which are currently assigned to the maximum number of users, are chosen to constitute set *S* in line 5. We merge the permissions of all the *k* roles into set $P_S$, and assign the $P_S$ to the newly created role while retaining the other ($MRC_{user}$ − 1) roles assigned to user *u* (lines 6–7). Then, the new role is inserted into the *PA* and *UA*. We update matrices *PA* and *UA*, based on the new role, in Lines 8–22. As a result, the UCC is satisfied, with a possible reduction in the number of roles assigned to the violating user, because the newly created role is used instead of the *k* merging roles.

### 3.2.2. Role Optimization Satisfying PCC

**Definition 5.** *(role-optimization problem with PCC) Given a user-permission assignments matrix $UPA_{n \times m}$, the preprocessed results UA and PA matrices, and a particular threshold $MRC_{user}$, find an optimal set R of roles, such that the UA and PA are consistent with the UPA, the number of roles to which any permission can be assigned is less than or equal to $MRC_{permission}$, and the number of the optimal roles is minimized. It can be formalized as follows:*

$$\begin{cases} \min |R| \\ UA \otimes PA = UPA \\ \sum_j PA[j][t] \leq MRC_{permission} \leq |R|, \forall t \in [1, m] \end{cases} \tag{17}$$

According to Definition 5, in order to satisfy role optimization in the presence of PCC, the optimizing process is presented in Algorithm 5.

---

**Algorithm 5.** Role optimization satisfying PCC.

---

**Input:** the preprocessed matrices *UA*, *PA*, initial role set *CR*, and threshold $MRC_{permission}$
**Output:** the optimized matrices *UA* and *PA*
1. Define and compute *count_permission_roles(p)* as the number of roles related to permission *p*;
2. Define and compute *count_role_permissions(r)* as the number of permissions assigned to role *r*;
3. **while** $\exists p \in P$: *count_perm_roles(p)*>$MRC_{permission}$ **do**
4.      *k* = *count_perm_roles(p)* − ($MRC_{permission}$ − 1);
5.      Choose the top *k* roles from *p* with the highest *count_role_permissions(r)* values to constitute set *S*;
6.      Intersect the permissions of all the *k* roles and denote the intersection as set $P_S$;
7.      Create a new role $r_{nr}$ such that *role_permissions*($r_{nr}$) = $P_S$;
8.      **for** each $u_i$ in *U* **do**
9.        **if** *count_user_roles*($u_i$)$\supseteq r_{nr}$ **then**
10.        *UA[i][nr]* = 1;
11.      **else**
12.        *UA[i][nr]* = 0;
13.      **end if**
14. **end for**
15.      **for** each $r_j$ in *S* **do**
16.        **if** $\forall p_t \in P_S$: *PA[j][t]* = = 1 **then**
17.        $\forall p_t \in P_S$: *PA[j][t]* = 0;
18.        *PA[nr][t]* = 1;
19.      **else**
20.        *PA[nr][t]* = 0;
21.      **end if**
22.      **end for**
23. Update *count_perm_roles(p)* and *count_role_perms (r)*;
24. **end while**

---

In Algorithm 5, we first define two functions: *Count_permission_roles(p)* and *count_role_permissions(r)* (lines 1–2). Line 3 determines whether the number of roles to which any permission is assigned exceeds $MRC_{permission}$. For each violating permission *p*, *k* number of roles is represented by calculating *count_permission_roles(p)* − ($MRC_{permission}$ − 1) in line 4, and the top *k* roles, which possess the maximum number of permissions, are chosen to constitute set *S* in line 5. We intersect the permissions of all the *k* roles into set $P_S$, and assign $P_S$ to the newly created role while retaining the other ($MRC_{permission}$ − 1) roles related to permission *p* (lines 6–7). Then, the new role is inserted into the *UA* and *PA*. We update matrices *UA* and *PA*, based on the new role, in Lines 8–22. As a result, the PCC is satisfied with a possible reduction in the number of roles related to the violating permission, because the newly created role is used instead of the *k* intersecting roles.

### 3.2.3. Role Optimization Satisfying both UCC and PCC

Algorithm 4 and Algorithm 5 show that either, the UCC or PCC, are considered. However, with an increase in the number of roles related to permission $p_t$ (owing to the merging roles), lines 9–10 may cause a violation of the PCC in Algorithm 4. Similarly, with an increase in the number of roles assigned to user $u_i$ (owing to the intersecting roles), lines 9–10 may cause a violation of the UCC in Algorithm 5. Thus, it is necessary to study whether the double constraints are satisfied simultaneously.

**Definition 6.** *(a role-optimization problem with both UCC and PCC) Given a user-permission assignment matrix $UPA_{n \times m}$, the preprocessed result matrices UA and PA, and two thresholds $MRC_{user}$ and $MRC_{permission}$, find an optimal set R of roles, such that the UA and PA are consistent with the UPA, where the number of roles assigned to any user is less than or equal to $MRC_{user}$, the number of roles to which any permission can be assigned is less than or equal to $MRC_{permission}$, and the number of the optimal roles is minimized. This can be formalized as follows:*

$$
\begin{cases}
\min |R| \\
UA \otimes PA = UPA \\
\sum_j UA[i][j] \leq MRC_{user} \leq |R|, \forall i \in [1, n] \\
\sum_j PA[j][t] \leq MRC_{permission} \leq |R|, \forall t \in [1, m]
\end{cases}
\tag{18}
$$

In addition, a role set *RU*, which would not cause any new violations in the role-permission assignments, needs to be identified; another role set *RI*, which would not cause any new violations in the user-role assignments, also needs to be identified. Specifically, for each role *r* in *RU*, any permission assigned to *r* does not violate the PCC when *r* is chosen and implemented in Algorithm 4. This can be represented as:

$\forall r \in RU, \forall p \in role\_permissions(r): count\_perm\_roles(p) \leq MRC_{permission} - 1.$

Similarly, for each role *r* in *RI*, any user possessing *r* would not violate the UCC when *r* is chosen and implemented in Algorithm 5. This process can be represented as:

$\forall r \in RI, \forall u \in role\_users(r): count\_user\_roles(u) \leq MRC_{user} - 1.$

According to Definition 6, the optimizing process is presented in Algorithm 6.

---

**Algorithm 6.** Role optimization satisfying both UCC and PCC.

---

**Input:** preprocessed matrices *UA*, *PA*, initial role set *CR*, and thresholds $MRC_{user}$ and $MRC_{permission}$
**Output:** optimized matrices *UA* and *PA*
1. Define and compute *count_user_roles(u), count_role_users(r), count_permission_roles(p), and count_role_permissions(r)*;
2. Identify *RU*, *RI*;
3. **while** ($\exists u \in U: count\_user\_roles(u) > MRC_{user}$) or
($\exists p \in P: count\_perm\_roles(p) > MRC_{permission}$) **do**
4.   Choose violating users or violating permissions based on a heuristic strategy;
5.    **if** user *u* is chosen **then**
6.     $k = count\_user\_roles(u) - (MRC_{user} - 1)$;
7.     Choose the top *k* roles of *u* from *RU* with the highest *count_role_users(r)* values to constitute set *S*;
8.     Merge the permissions of all the *k* roles and denote the union as set $P_S$;
9.     Create a new role $r_{nr}$ such that *role_permissions*($r_{nr}$) = $P_S$;
10.    Update the *PA* and *UA* with $r_{nr}$ according to Algorithm 4;
11.   **else**
12.    $k = count\_perm\_roles(p) - (MRC_{permission} - 1)$;
13.    Choose the top *k* roles of *p* from *RI* with the highest *count_role_permissions(r)* values to constitute set *S*;
14.    Intersect the permissions of all the *k* roles and denote the intersection as set $P_S$;
15.    Create a new role $r_{nr}$ such that *role_permissions*($r_{nr}$) = $P_S$;
16.    Update the *UA* and *PA* with $r_{nr}$ according to Algorithm 5;
17.   **end if**
18. **end while**

---

In Algorithm 6, we first identify *RU*, *RI*, and determine the violating users or permissions using a heuristic strategy in lines 2–4. Then, if user *u* is chosen, the top *k* roles are chosen from *RU*; if permission *p* is chosen, the top *k* roles are chosen from *RI*. Similar to updating the *UA* and *PA* in Algorithm 4 and Algorithm 5, detailed descriptions of the algorithm are omitted, due to the limited space.

## 3.3. Role Assignments Satisfying Multiple Constraints

In this subsection, we study the issue of assigning the available users with different capabilities to the mining roles in the role-engineering system, such that the number of user-role assignments is maximized while satisfying the relevant security constraints. Specifically, we study how to assign

each role with the maximum number of users based on the two conditions: (1) Retaining the UCC in role-mining optimization, and (2) considering the user-oriented mutually exclusive constraints and constraint degree of the roles in role assignments. First, we present the following definitions.

**Definition 7.** *(A user-oriented mutually exclusive constraint) Given a matrix UC for user-capability constraints, consider a constructed set C for the t-t SMER constraints in the role-engineering system. Then, the role assignments for any user in UC should satisfy both the UC and C constraints.*

**Definition 8.** *(The constraint degree of a role) Let the t-t SMER constraint set be C = {$c_1,c_2, \ldots c_i, \ldots$ }, where $c_i = smer<\{r_1,r_2, \ldots r_{ti}\},ti>$. The percentage of different constraints, including role r, in C is called the constraint degree of r with respect to C, which is represented as,*

$$smer_C(r) = \frac{|\{c_k|\exists c_k \in C : r \text{ is included in } c_k\}|}{|C|} \qquad (19)$$

*where $smer_C(r) \in (0,1]$.*

**Definition 9.** *(A role-assignment problem with multiple constraints) Besides the set U of users, set R of roles, and threshold $MRC_{user}$ for the UCC constraint in the role-mining optimization, given the matrix UC for user-capability constraints, and a set C for the t-t SMER constraints, we find a role-assignment matrix UA' such that the number of role assignments is maximized while satisfying all the constraints. This matrix can be formalized as follows:*

$$\begin{cases} \max|UA'| \\ UA'[i][j] = 0, \forall UC[i][j] = 0 \\ user\_roles(u) \ satisfy \ C, \forall u \in UA' \\ count\_user\_roles(u) \leq MRC_{user}, \forall u \in UA' \end{cases} \qquad (20)$$

According to Definitions 7–9, the assigning process is presented in Algorithm 7.

Obviously, it is observed that if user $u_i$ in *UC* cannot perform role $r_j$, then we say that $(u_i, r_j) \notin UA'$; otherwise, it is uncertain whether $r_j$ can be assigned to $u_i$. On the other hand, if $(u_i, r_j) \in UA'$, then we say that $(u_i, r_j) \in UC$; otherwise, it is uncertain whether $u_i$ can or cannot perform $r_j$. We formalize these observations as follows:

(1) $\forall i, \forall j : UC[i][j] == 0 \Rightarrow UA'[i][j] == 0, UC[i][j] == 1 \nRightarrow UA'[i][j] == 1$;
(2) $\forall i, \forall j : UA'[i][j] == 1 \Rightarrow UC[i][j] == 1, UA'[i][j] == 0 \nRightarrow UC[i][j] == 0$.

In Algorithm 7, we first initialize *UA'*, based on the observations in lines 1–7, and represent the uncertainty with variable $a_{ij}$ (line 5). Then, we create a new priority queue *Q* and insert set *R* into *Q*, according to the constraint degree of roles, where the lower the $smer_C(r)$ value, the higher the priority of role *r* (lines 8–9). Next, we implement the role assignments while satisfying all the security constraints until *Q* is empty (lines 10–22).

---

**Algorithm 7.** Role assignments satisfying multiple constraints.

---

**Input:** Set $U$ for users, set $R$ for roles, threshold $MRC_{user}$, matrix $UC$ for user-capability constraints, *and* set $C$ for *t-t* SMER constraints

**Output:** user-role assignment matrix $UA'$

1. **for** each $UC[i][j]$ in $UC$ **do**
2.   **if** $UC[i][j] == 0$ **then**
3.     $UA'[i][j] = 0$;
4.   **else**
5.     $UA'[i][j] = a_{ij}$;
6.   **end if**
7. **end for**
8. Create a new priority queue $Q$ and insert all roles of $R$ into $Q$;
9. Sort roles in $Q$ according to the ascending order of their constraint degree. Role $r$, which has a lower $smer_C(r)$ value, has a higher priority;
10. **while** $Q$ is not empty **do**
11.   Choose role $r_j$ with the highest priority in $Q$;
12.   **for** each $u_i$ in $UC$ **do**
13.     **if** $UA'[i][j] \neq 0$ **then**
14.     **for** each $smer<\{r_1, r_2, \ldots, r_t\}, t>$ in $C$ **do**
15.      **if** $(|\{user\_roles(u_i) \cup \{r_j\}\} \cap \{r_1, r_2, \ldots, r_t\}| < t)$ and $(count\_user\_roles(u_i) \leq MRC_{user} - 1)$ **then**
16.       $UA'[i][j] = 1$;
17.      **end if**
18.     **end for**
19.    **end if**
20.   **end for**
21.   Remove $r_j$ from $Q$;
22. **end while**

---

## 4. Theoretical Analyses and Running Examples

### 4.1. Relationship between the Center Point and Compression Point

According to Algorithm 2, since user cluster *CU* is divided into *k* partitions, *k* different compression points need to be created in most of the compression processes. The storage space for many compression points becomes much larger with an increasing number of user clusters. In order to reduce the cost of storage, it is necessary to find suitable substitutes from existing users to meet the characteristics of the compression points.

**Statement 1.** *The center point of any partition can be a substitute for the compression point in line 5 of Algorithm 3.*

**Proof.** We analyze the statement from two perspectives: Sufficiency and necessity.

Sufficiency: Definition 1 indicates that the average similarity of the center point to all the non-center points in any partition is maximal. Meanwhile, it is observed from equation (11) that the number of users who possess permission *p* assigned to center point $u_i$ is no less than that of permission $p'$ assigned to any non-center point. That is,

$$\forall p \in user\_permissions(u_i), \forall p' \notin user\_permissions(u_i):$$
$$\left|\{u|\exists u \in associate(u_i):(u,p) \in UPA\}\right| \geq \left|\{u'|\exists u' \in associate(u_i):(u',p') \in UPA\}\right|$$
$$\Rightarrow support_{associate(u_i)\cup\{u_i\}}(p) \geq support_{associate(u_i)\cup\{u_i\}}(p') \geq t.$$

This is clearly true.

Necessity: We use the method of contradiction and assume that the support degree of permission $p$, which is assigned to center point $u_i$, does not satisfy the decision condition in Algorithm 3. In other words, there exists a satisfied permission $p'$ not assigned to $u_i$. That is,

$\exists p \in assigned\_permissions(u_i), \exists p' \notin assigned\_permissions(u_i):$
$support_{associate(u_i) \cup \{u_i\}}(p') \geq t > support_{associate(u_i) \cup \{u_i\}}(p)$
$\Rightarrow \left| \{u' | \exists u' \in associate(u_i) : (u', p') \in UPA\} \right| \geq \left| \{u | \exists u \in associate(u_i) : (u, p) \in UPA\} \right|.$

Then,

$\exists u_j \in \{u' | \exists u' \in associate(u_i) : (u', p') \in UPA\}:$
$sim(u_j, associate(u_i) \backslash \{u_j\} \cup \{u_i\}) > sim(u_i, associate(u_i))$
$\Rightarrow dis(u_j, associate(u_i) \backslash \{u_j\} \cup \{u_i\}) < dis(u_i, associate(u_i)).$

It is contradictory to the case that $u_i$ is the center point. Thus, the assumption is false.  □

*4.2. The Influencing Factors of Role Assignments*

According to Definition 9, the task of the role assignment aims to find matrix $UA'$ such that the number of role assignments is maximized. In order to analyze the assigning efficiency of the method, we present the definition of the role-utilization ratio, as follows:

**Definition 10.** *(role-utilization ratio, RR) Given matrix UA' of the role assignments and matrix UC of the user-capability constraints, the RR is calculated as follows,*

$$RR = \frac{|UA'|}{|UC|} \times 100\% \tag{21}$$

*where $RR \in (0, 1]$.*

Indeed, $RR$ is the percentage of the number of 1 in the $UA'$ compared to that in the $UC$. Note that if none of the constraints are considered, then $RR$ is equal to 100% because $UA' = UC$. However, $RR$ is influenced by various parameters during the whole process of the role-engineering optimization. The value of $RR$ varies with the varying truth assignments for the $UA'$. Thus, it is necessary to study how $RR$ is influenced by existing factors, such as cardinality constraints and user-oriented mutually exclusive constraints.

**Statement 2.** *Given the threshold $MRC_{user}$ of the UCC constraint, set R' of the roles in the UC constraint, and the density $\delta$ of the UC matrix (that is, a percentage of 1 in the UC), the upper bound of RR satisfies the following:*

$$RR < \frac{MRC_{user}}{|R'| \times \delta}.$$

**Proof.** The number of cells in matrix $UC$ is $|U'| \times |R'|$. As the percentage of 1 in the $UC$ is $\delta$, $|UC| = |U'| \times |R'| \times \delta$. Meanwhile, as any user in the $UC$ is *assigned $MRC_{user}$* roles, at most, $|UA'| < |U'| \times MRC_{user}$. Thus, $RR = \frac{|UA'|}{|UC|} < \frac{|U'| \times MRC_{user}}{|U'| \times |R'| \times \delta} = \frac{MRC_{user}}{|R'| \times \delta}$.  □

It is observed from Statement 2 that the value of $RR$ increases with an increasing value of $MRC_{user}$, and decreases as the number of roles and the density of the $UC$ increase. Furthermore, it is observed from line 14 of Algorithm 7 that $RR$ is also influenced by the number of constraints in set $C$. As the truth assignments for the $UA'$ decrease with an increasing number of the constraints, the value of $RR$ decreases with an increasing value of $|C|$.

### 4.3. Relationship between the UCC and PCC

Although, both optimized results can satisfy a single constraint requirement, according to Algorithm 4 and Algorithm 5, another security constraint may be violated. In other words, they do not address the issue of balancing the role-mining effectiveness and system security.

**Statement 3.** *The UCC and PCC are mutually exclusive.*

**Proof.** We can analyze the statement from lines 9–10 in Algorithm 4 and lines 9–10 in Algorithm 5. However, detailed descriptions are omitted due to limited space. That is, it is a contradictory relationship between the UCC and PCC.  □

### 4.4. Running Examples

In order to eliminate redundancies while maintaining the system's usability for role mining, we handle the original permission assignments $UPA_{original}$ using the compression technology and convert the uncompressed matrix $UPA_{original}$ into the compressed matrix $UPA_{compressed}$. Note that the $UPA_{compressed}$ is much denser than the $UPA_{original}$, particularly in large-scale access control systems. Indeed, it is convenient and feasible to analyze and handle the compressed data object. As shown in Table 2, $(UPA_{compressed})_{6\times6}$ is a compressed matrix, where the shadow parts are dense and provide motivation for available mining roles. The following example is presented to demonstrate the effectiveness of our method.

**Table 2.** Compressed matrix $(UPA_{compressed})_{6\times6}$.

| 1 | 1 | 0 | 0 | 0 | 0 |
|---|---|---|---|---|---|
| 1 | 1 | 0 | 0 | 0 | 0 |
| 0 | 0 | 1 | 1 | 1 | 0 |
| 0 | 0 | 1 | 1 | 1 | 1 |
| 0 | 0 | 0 | 1 | 1 | 1 |
| 0 | 0 | 0 | 1 | 1 | 1 |

**Example 1.** *This example considers the matrix $UPA_{original}$ of the original permission assignments, a k number of center points, a threshold t of the support degree, a threshold $MRC_{user}$ of the UCC, and a threshold $MRC_{permission}$ of the PCC, where the $UPA_{original}$ is comprised of 15 users and 6 permissions, as shown in Table 3, k = 2, t = 0.66, and $MRC_{user} = MRC_{permission} = 2$.*

In the preprocessing phase, we first calculate the similarities between different users based on equation (11) and the results of Algorithm 1 and identify user clusters $CU = \{CU_1, CU_2\}$, where $CU_1 = \{u_1, u_2, u_4, u_5, u_{11}, u_{12}, u_{13}, u_{14}\}$ and $CU_2 = \{u_3, u_6, u_7, u_8, u_9, u_{10}, u_{15}\}$. Then, we use the partitioning technology (Algorithm 2) to handle each user cluster with k = 2 independently. As shown in the results in Table 4, $CU_1$ is divided into two partitions: $\{u_1, u_2, u_4, u_{11}\}$ (with center point $u_4$) and $\{u_5, u_{12}, u_{13}, u_{14}\}$ (with center point $u_{12}$). Similarly, $CU_2$ is also divided into two partitions, as shown in Table 5. Next, we use the compression technology (Algorithm 3) to compress each partition with t = 0.66. As shown in Tables 4 and 5, the full-line shadow parts, which satisfy the compressing condition, are regarded as usable for mining roles. The dotted-line shadow fractions, such as $(u_1, p_6)$, $(u_{11}, p_5)$, and $(u_5, p_6)$, which do not satisfy the compressing condition, are regarded as redundant. The compressed matrix is shown in Table 6, and the set of mining roles is $R_{initial} = \{\{p_1, p_2, p_4\}, \{p_2, p_3, p_4\}, \{p_2, p_4\}, \{p_4\}, \{p_3, p_6\}, \{p_5, p_6\}, \{p_6\}\}$.

In the role optimization phase, we use Algorithm 6 with $MRC_{user} = MRC_{permission} = 2$, and the set of the optimized roles is $R_{optimized} = \{\{p_1, p_2\}, \{p_2, p_4\}, \{p_4\}, \{p_3\}, \{p_6\}, \{p_5\}\}$. The optimized results, which are shown in Table 7 and in Table 8, can satisfy the double constraints simultaneously.

**Table 3.** Original matrix $UPA_{original}$.

|  | $p_1$ | $p_2$ | $p_3$ | $p_4$ | $p_5$ | $p_6$ |
|---|---|---|---|---|---|---|
| $u_1$ | 0 | 0 | 0 | 1 | 0 | 1 |
| $u_2$ | 1 | 1 | 0 | 1 | 0 | 0 |
| $u_3$ | 0 | 0 | 0 | 0 | 1 | 1 |
| $u_4$ | 1 | 1 | 0 | 1 | 0 | 0 |
| $u_5$ | 0 | 1 | 1 | 1 | 0 | 1 |
| $u_6$ | 0 | 0 | 0 | 0 | 1 | 1 |
| $u_7$ | 0 | 0 | 1 | 0 | 0 | 1 |
| $u_8$ | 0 | 1 | 1 | 0 | 0 | 1 |
| $u_9$ | 1 | 0 | 1 | 0 | 0 | 1 |
| $u_{10}$ | 0 | 0 | 0 | 1 | 1 | 0 |
| $u_{11}$ | 1 | 1 | 0 | 1 | 1 | 0 |
| $u_{12}$ | 0 | 1 | 1 | 1 | 0 | 0 |
| $u_{13}$ | 0 | 1 | 1 | 1 | 0 | 0 |
| $u_{14}$ | 0 | 1 | 0 | 1 | 0 | 0 |
| $u_{15}$ | 1 | 0 | 1 | 0 | 0 | 1 |

**Table 4.** User cluster $CU_1$.

|  | $p_1$ | $p_2$ | $p_4$ | $p_3$ | $p_6$ | $p_5$ |
|---|---|---|---|---|---|---|
| $u_1$ | 0 | 0 | 1 | 0 | 1 | 0 |
| $u_2$ | 1 | 1 | 1 | 0 | 0 | 0 |
| *$u_4$ | 1 | 1 | 1 | 0 | 0 | 0 |
| $u_{11}$ | 1 | 1 | 1 | 0 | 0 | 1 |
| $u_5$ | 0 | 1 | 1 | 1 | 1 | 0 |
| *$u_{12}$ | 0 | 1 | 1 | 1 | 0 | 0 |
| $u_{13}$ | 0 | 1 | 1 | 1 | 0 | 0 |
| $u_{14}$ | 0 | 1 | 1 | 0 | 0 | 0 |

**Table 5.** User cluster CU$_2$.

|  | $p_4$ | $p_2$ | $p_1$ | $p_3$ | $p_6$ | $p_5$ |
|---|---|---|---|---|---|---|
| $u_{10}$ | 1 | 0 | 0 | 0 | 0 | 1 |
| *$u_3$ | 0 | 0 | 0 | 0 | 1 | 1 |
| $u_6$ | 0 | 0 | 0 | 0 | 1 | 1 |
| *$u_7$ | 0 | 0 | 0 | 1 | 1 | 0 |
| $u_8$ | 0 | 1 | 0 | 1 | 1 | 0 |
| $u_9$ | 0 | 0 | 1 | 1 | 1 | 0 |
| $u_{15}$ | 0 | 0 | 1 | 1 | 1 | 0 |

**Table 6.** Compressed matrix.

|       | $p_1$ | $p_2$ | $p_4$ | $p_3$ | $p_6$ | $p_5$ |
|-------|-------|-------|-------|-------|-------|-------|
| $u_1$ | 0 | 0 | **1** | 0 | 0 | 0 |
| $u_4$ | **1** | **1** | **1** | 0 | 0 | 0 |
| $u_{12}$ | 0 | **1** | **1** | **1** | 0 | 0 |
| $u_{14}$ | 0 | **1** | **1** | 0 | 0 | 0 |
| $u_3$ | 0 | 0 | 0 | 0 | **1** | **1** |
| $u_7$ | 0 | 0 | 0 | **1** | **1** | 0 |

**Table 7.** Optimized *UA*.

|       | $r_1$ | $r_2$ | $r_3$ | $r_4$ | $r_5$ | $r_6$ |
|-------|-------|-------|-------|-------|-------|-------|
| $u_1$ | 0 | 0 | 1 | 0 | 0 | 0 |
| $u_4$ | 1 | 0 | 1 | 0 | 0 | 0 |
| $u_{12}$ | 0 | 1 | 0 | 1 | 0 | 0 |
| $u_{14}$ | 0 | 1 | 0 | 0 | 0 | 0 |
| $u_3$ | 0 | 0 | 0 | 0 | 1 | 1 |
| $u_7$ | 0 | 0 | 0 | 1 | 1 | 0 |

**Table 8.** Optimized *PA*.

|       | $p_1$ | $p_2$ | $p_4$ | $p_3$ | $p_6$ | $p_5$ |
|-------|-------|-------|-------|-------|-------|-------|
| $r_1$ | 1 | 1 | 0 | 0 | 0 | 0 |
| $r_2$ | 0 | 1 | 1 | 0 | 0 | 0 |
| $r_3$ | 0 | 0 | 1 | 0 | 0 | 0 |
| $r_4$ | 0 | 0 | 0 | 1 | 0 | 0 |
| $r_5$ | 0 | 0 | 0 | 0 | 1 | 0 |
| $r_6$ | 0 | 0 | 0 | 0 | 0 | 1 |

To further demonstrate the effectiveness of our method, we simulate an actual-application scenario and implement role assignments in the role-engineering system. Table 9 presents descriptions of several roles used in the following example.

**Example 2.** *Besides the threshold $MRC_{user}$ of the UCC and the $R_{optimized}$ set of the optimized roles in Example 1, consider the following user-oriented mutually exclusive constraints:*

*(1)    The matrix UC of user-capability constraints, which is shown in Table 10;*

*(2)    set $C = \{c_1,c_2,c_3,c_4\}$ for the t-t SMER constraints, where $c_1 = smer<\{r_1,r_3\},2>$, $c_2 = smer<\{r_2,r_3\},2>$, $c_3 = smer<\{r_1,r_2,r_3\},3>$, and $c_4 = smer<\{r_4,r_5\},2>$.*

In the role assignment phase, we first initialize the *UA'* according to Algorithm 7, as shown in Table 11. Then, we determine the constraint degree of the different roles, with respect to set $C:smer_C(r_1)$ $= smer_C(r_2) = 0.5$, $smer_C(r_3) = 0.75$, $smer_C(r_4) = smer_C(r_5) = 0.25$. Then we insert set $\{r_1,r_2,r_3,r_4,r_5\}$ into queue $Q$ according to the constraint degree of the roles. Next, we implement the role assignments, as follows.

**Table 9.** Descriptions of roles.

| Role | Description |
|------|-------------|
| $r_1$ | Software Designer |
| $r_2$ | Software Developer |
| $r_3$ | Software Tester |
| $r_4$ | Accounts Manager |
| $r_5$ | Financial Auditor |

**Table 10.** Matrix *UC*.

| | $r_1$ | $r_2$ | $r_3$ | $r_4$ | $r_5$ |
|------|------|------|------|------|------|
| $u_1$ | 1 | 0 | 0 | 1 | 0 |
| $u_2$ | 0 | 0 | 1 | 1 | 1 |
| $u_3$ | 1 | 1 | 1 | 0 | 0 |
| $u_4$ | 0 | 0 | 0 | 1 | 1 |
| $u_5$ | 1 | 0 | 0 | 0 | 1 |
| $u_6$ | 1 | 1 | 1 | 1 | 1 |

**Table 11.** Initialization for matrix *UA'*.

| | $r_1$ | $r_2$ | $r_3$ | $r_4$ | $r_5$ |
|------|------|------|------|------|------|
| $u_1$ | $a_{11}$ | 0 | 0 | $a_{14}$ | 0 |
| $u_2$ | 0 | 0 | $a_{23}$ | $a_{24}$ | $a_{25}$ |
| $u_3$ | $a_{31}$ | $a_{32}$ | $a_{33}$ | 0 | 0 |
| $u_4$ | 0 | 0 | 0 | $a_{44}$ | $a_{45}$ |
| $u_5$ | $a_{51}$ | 0 | 0 | 0 | $a_{55}$ |
| $u_6$ | $a_{61}$ | $a_{62}$ | $a_{63}$ | $a_{64}$ | $a_{65}$ |

Here, $r_4$ is chosen as the candidate role according to its priority in step 1, and we set $a_{14} = a_{24} = a_{44} = a_{64} = 1$, $a_{25} = a_{45} = a_{65} = 0$, which is shown in Table 12; $r_5$ is chosen as the candidate role in step 2, and we can set $a_{55} = 1$, which is shown in Table 13; $r_1$ is chosen as the candidate role in step 3, and we set $a_{11} = a_{31} = a_{51} = a_{61} = 1$, $a_{33} = a_{63} = 0$, which is shown in Table 14; $r_2$ is chosen as the candidate role in step 4, we set $a_{32} = 1$, and $a_{62}$ is 0 because of the cardinality constraint, which is shown in Table 15; lastly, $r_3$ is chosen as the candidate role, and we set $a_{23} = 1$. As per the results shown in Table 16, the role assignments satisfy all the given constraints. Table 17 presents the assigning process that does not stop until queue $Q$ becomes empty.

**Table 12.** Assignments in step 1.

| | $r_1$ | $r_2$ | $r_3$ | $r_4$ | $r_5$ |
|------|------|------|------|------|------|
| $u_1$ | $a_{11}$ | 0 | 0 | **1** | 0 |
| $u_2$ | 0 | 0 | $a_{23}$ | **1** | **0** |
| $u_3$ | $a_{31}$ | $a_{32}$ | $a_{33}$ | 0 | 0 |
| $u_4$ | 0 | 0 | 0 | **1** | **0** |
| $u_5$ | $a_{51}$ | 0 | 0 | 0 | $a_{55}$ |
| $u_6$ | $a_{61}$ | $a_{62}$ | $a_{63}$ | **1** | **0** |

**Table 13.** Assignments in step 2.

|  | $r_1$ | $r_2$ | $r_3$ | $r_4$ | $r_5$ |
|---|---|---|---|---|---|
| $u_1$ | $a_{11}$ | 0 | 0 | **1** | 0 |
| $u_2$ | 0 | 0 | $a_{23}$ | **1** | **0** |
| $u_3$ | $a_{31}$ | $a_{32}$ | $a_{33}$ | 0 | 0 |
| $u_4$ | 0 | 0 | 0 | **1** | **0** |
| $u_5$ | $a_{51}$ | 0 | 0 | 0 | **1** |
| $u_6$ | $a_{61}$ | $a_{62}$ | $a_{63}$ | **1** | **0** |

**Table 14.** Assignments in step 3.

|  | $r_1$ | $r_2$ | $r_3$ | $r_4$ | $r_5$ |
|---|---|---|---|---|---|
| $u_1$ | **1** | 0 | 0 | **1** | 0 |
| $u_2$ | 0 | 0 | $a_{23}$ | **1** | **0** |
| $u_3$ | **1** | $a_{32}$ | **0** | 0 | 0 |
| $u_4$ | 0 | 0 | 0 | **1** | **0** |
| $u_5$ | **1** | 0 | 0 | 0 | **1** |
| $u_6$ | **1** | $a_{62}$ | **0** | **1** | **0** |

**Table 15.** Assignments in step 4.

|  | $r_1$ | $r_2$ | $r_3$ | $r_4$ | $r_5$ |
|---|---|---|---|---|---|
| $u_1$ | **1** | 0 | 0 | **1** | 0 |
| $u_2$ | 0 | 0 | $a_{23}$ | **1** | **0** |
| $u_3$ | **1** | **1** | **0** | 0 | 0 |
| $u_4$ | 0 | 0 | 0 | **1** | **0** |
| $u_5$ | **1** | 0 | 0 | 0 | **1** |
| $u_6$ | **1** | **0** | **0** | **1** | **0** |

**Table 16.** Assignments in step 5.

|  | $r_1$ | $r_2$ | $r_3$ | $r_4$ | $r_5$ |
|---|---|---|---|---|---|
| $u_1$ | **1** | 0 | 0 | **1** | 0 |
| $u_2$ | 0 | 0 | **1** | **1** | **0** |
| $u_3$ | **1** | **1** | **0** | 0 | 0 |
| $u_4$ | 0 | 0 | 0 | **1** | **0** |
| $u_5$ | **1** | 0 | 0 | 0 | **1** |
| $u_6$ | **1** | **0** | **0** | **1** | **0** |

**Table 17.** The assigning process.

| Step | Candidate Role | Identified $a_{ij}$ | Assigned Users | Updated $Q$ |
|---|---|---|---|---|
| 1 | $r_4$ | $a_{14}, a_{24}, a_{44}, a_{64}, a_{25}, a_{45}, a_{65}$ | $u_1, u_2, u_4, u_6$ | $\{r_5, r_1, r_2, r_3\}$ |
| 2 | $r_5$ | $a_{55}$ | $u_5$ | $\{r_1, r_2, r_3\}$ |
| 3 | $r_1$ | $a_{11}, a_{31}, a_{51}, a_{61}, a_{33}, a_{63}$ | $u_1, u_3, u_5, u_6$ | $\{r_2, r_3\}$ |
| 4 | $r_2$ | $a_{32}, a_{62}$ | $u_3$ | $\{r_3\}$ |
| 5 (finish) | $r_3$ | $a_{23}$ | $u_2$ | $\Phi$ |

## 5. Experimental Evaluations

In this section, three groups of experiments are carried out. The first group of experiments is used to evaluate the accuracy of REO_CCUMEC, the second is to evaluate its effectiveness, and the third is to evaluate its efficiency. We comprehensively consider 11 datasets from the work in [14]. These datasets are both real and synthetic, and the regular mining tool RMiner [48] was run on these datasets in the literature to evaluate the performance of unconstrained role mining. The original datasets, including the density of the dataset, the number of the initial role set *CR*, and the execution time, are shown in Table 18. All experiments are implemented on a standard desktop PC with an Intel i5–7400 CPU, 4 GB RAM, and 160 GB hard disks, running a 64-bit Windows 7 operating system. All simulations are compiled and executed in Eclipse IDE for a Java Developer environment.

**Table 18.** Original datasets.

| Dataset | $|U|$ | $|P|$ | $|UPA|$ | Density | $|CR|$ | Execution Time(s) |
|---|---|---|---|---|---|---|
| America-large | 3485 | 10,127 | 185,294 | 0.5% | 423 | 78.78 |
| America-small | 3477 | 1587 | 105,205 | 1.9% | 213 | 6.31 |
| Apj | 2044 | 1164 | 6841 | 0.3% | 456 | 5.60 |
| Customer | 10,961 | 284 | 45,427 | 1.5% | 276 | 4.66 |
| Domino | 79 | 231 | 730 | 4% | 20 | 0.01 |
| Emea | 35 | 3046 | 7,20 | 6.8% | 34 | 0.02 |
| Firewall1 | 365 | 709 | 31,951 | 12.3% | 69 | 0.11 |
| Firewall2 | 325 | 590 | 36,428 | 19% | 10 | 0.15 |
| Healthcare | 46 | 46 | 1486 | 70% | 15 | 0.01 |
| University1 | 493 | 56 | 3955 | 14.3% | 31 | 0.01 |
| University2 | 400 | 14 | 3073 | 54.9% | 15 | 0.01 |

### 5.1. The Accuracy of the REO_CCUMEC

#### 5.1.1. Experimental Setup

We denote the percentage of the number of center points in the user cluster as the *compression ratio*. For the given user cluster *CU* with *k* center points, the *compression ratio* $= k/|CU|$. The experimental setup includes to following. The *compression ratio* increases from 0.05 to 0.4 with a step of 0.05, and we choose 0.66, 0.8, 1 as the threshold *t* of the support degree. In addition, the partitioning and compression algorithms are written in Java.

#### 5.1.2. Evaluation Measures

To evaluate the accuracy of the REO_CCUMEC in the preprocessing phase, on the one hand, we consider the similarity between the roles from the compressed results and initial roles with respect to the same set of users as one measure, which is denoted as $sim_U(UA_{compressed}, UA_{initial})$; on the other hand, we consider the similarity with respect to the same set of permissions as another measure, which is denoted as $sim_P(PA_{compressed}, PA_{initial})$.

#### 5.1.3. Experimental Results and Analyses

We implement the experiments on the America-large, America-small, University1, and University2 datasets, as shown in Table 18, and take the median value. The results of the experiments are shown in Figures 1 and 2, where the lateral axis represents the varying values of the *compression ratio*, and the vertical axis represents the changes of similarity.

Figure 1 demonstrates that the value of $sim_P(PA_{compressed}, PA_{initial})$ does not obviously vary as the *compression ratio* increases under different *t* values. This value is very smooth and always remains above 0.95; no matter how one divides the user cluster into partitions, permissions are inserted into the compressed matrix once their support degree exceeds the threshold *t*, according to Algorithm

3. For the same permission set, the roles from the compressed results are the same as the initial ones. Although, there remains the possibility of temporary changes in access-resource permissions in large-scale application systems, a dissimilarity below 0.05 can be accepted. Therefore, it is accurate to use the method of preprocessing from the viewpoint of $sim_P(PA_{compressed}, PA_{initial})$.

Figure 2 demonstrates that the values of $sim_U(UA_{compressed}, UA_{initial})$ are different under different $t$ values. The value of $sim_U(UA_{compressed}, UA_{initial})$ increases slightly as the *compression ratio* increases when $t = 0.66$, but it tends to grow linearly as the *compression ratio* increases when $t = 0.8$ or 1. The higher the *compression ratio*, the greater the number of compression points (that is, the center point) and the roles assigned to the users. In addition, The value of $sim_U(UA_{compressed}, UA_{initial})$ is no lower than 0.6 as the *compression ratio* increases when $t = 0.66$ but remains less than 0.6 when the *compression ratio* grows to 0.25 when $t = 0.8$ or 1. The higher the threshold $t$ of the support degree, the fewer the users and the roles that satisfy the corresponding requirements. Therefore, the results are less accurate from the viewpoint of $sim_U(UA_{compressed}, UA_{initial})$ when $t$ exceeds 0.66.

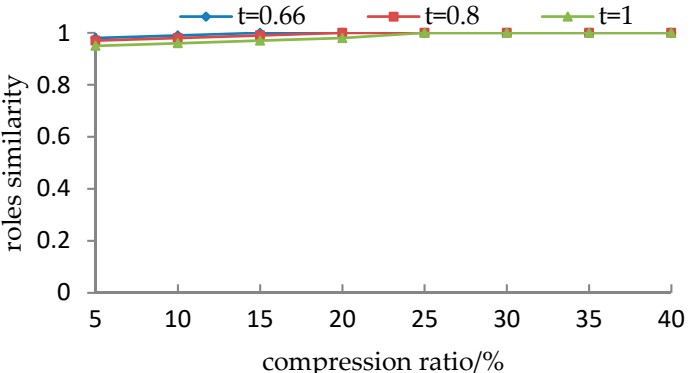

**Figure 1.** Comparison of $sim_P(PA_{compressed}, PA_{initial})$.

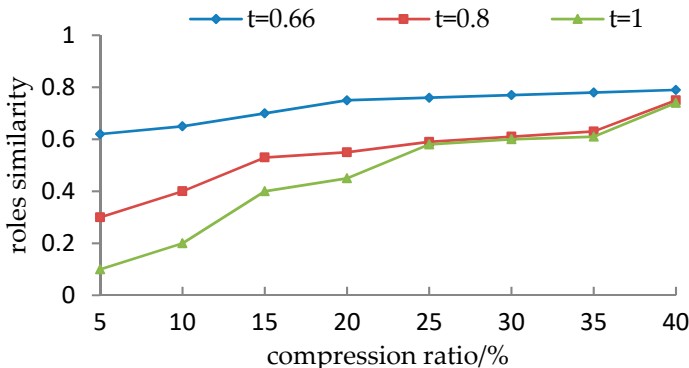

**Figure 2.** Comparison of $sim_U(UA_{compressed}, UA_{initial})$.

## 5.2. The Effectiveness of the REO_CCUMEC

### 5.2.1. Performance Evaluations under a Single Constraint

We first study how the number of optimized roles is influenced by a single cardinality constraint according to Algorithms 4–5. The preprocessing results *UA* and *CR* are considered to be inputs; both thresholds, $MRC_{user}$ and $MRC_{permission}$, are greater than 1, and we implement the experiments on the Domino and Healthcare datasets, as shown in Table 18.

In order to evaluate the effectiveness of the REO_CCUMEC under the UCC, we compare the performance of our method with the results of the representative RPA and CPA [38]. The results are

shown in Figures 3 and 4, where the lateral axis represents varying values of the threshold *MRCuser*, and the vertical axis represents varying values for the number of roles.

It is observed from Figure 3 that the number of roles decreases as $MRC_{user}$ increases in the REO_CCUMEC, which tends to be stable and no longer changes as $MRC_{user}$ increases to a certain value. Specifically, the number of roles does not obviously vary and remains close to 20 when the value of $MRC_{user}$ exceeds 8. Note that the number of the initial mining roles from the Domino dataset is 20 as shown in Table 18. Thus, the maximum number of roles assigned to any user can be regarded as 8 in the case of unconstrained mining. A further observation is that the number of roles first varies slightly and then increases significantly as $MRC_{user}$ decreases. The reason for this result is that the greater the value of $MRC_{user}$, the more roles assigned to any user (that is, not too many permissions need to be assigned to a regular role) and the weaker the constraint. In other words, with a greater value of $MRC_{user}$, regular roles are more applicable and can be utilized more frequently. Thus, fewer irregular roles need to be created, and the number of roles does not vary considerably. On the contrary, the smaller the value of $MRC_{user}$, the stronger the constraint. More permissions are assigned to irregular roles that are rarely utilized, and the number of roles increases remarkably because of the creation of more new roles.

However, the number of roles tends to increase as $MRC_{user}$ increases, from 1 to 4, in both the RPA and CPA, which seems to be contradictory. The reason for this result is that the Domino dataset contains exclusive permissions and produces exclusive roles in the presence of constraints. As shown in the figure, the maximum number of roles is close to 30 when $MRC_{user}$ equals 4, while the minimum number of roles is 23 when $MRC_{user}$ equals 1. Therefore, our method outperforms the RPA and CPA in the Domino dataset.

Similar to the analyses in Figure 3, it is observed in Figure 4 that the number of roles also decreases as $MRC_{user}$ increases in the REO_CCUMEC, which tends to be stable and remains close to 15 when $MRC_{user}$ increases to a certain value. However, the variations of the results in both the RPA and CPA are simple. The RPA generates 15 roles that remain unchanged when $MRC_{user}$ exceeds 1, while the number of roles is 18 when $MRC_{user}$ equals 1; the CPA generates 18 roles that remain unchanged as $MRC_{user}$ varies. Therefore, our method outperforms the CPA in the Healthcare dataset.

In order to evaluate the effectiveness of the REO_CCUMEC under the PCC, a representative post-processing method [39] is used for the performance comparison. The results are shown in Figures 5 and 6, where the lateral axis represents the varying values of the threshold $MRC_{permission}$, and the vertical axis represents the varying values of the number of roles. It is observed that the number of roles first varies slightly as $MRC_{permission}$ decreases and then increases significantly in the two methods. Moreover, the number of optimized roles in our method is close to that in the post-processing method. Detailed analyses are not discussed in this paper as similar discussions have been presented for the changes of $MRC_{user}$ values. Therefore, our method is as effective as the post-processing method in the Domino and Healthcare datasets.

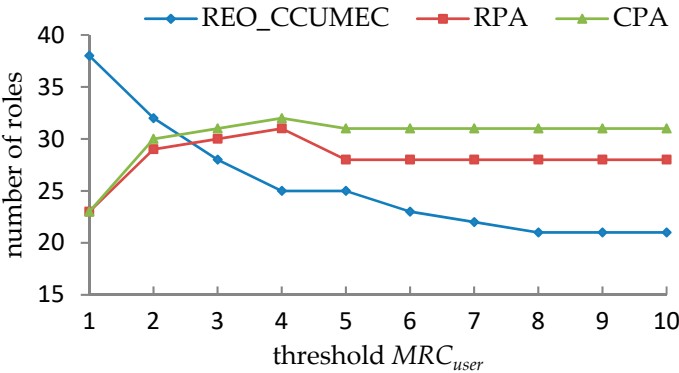

**Figure 3.** Performance comparison under the UCC in Domino.

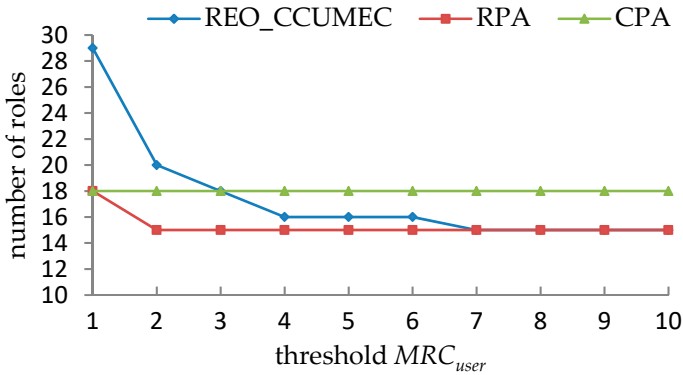

**Figure 4.** Performance comparison under the UCC in Healthcare.

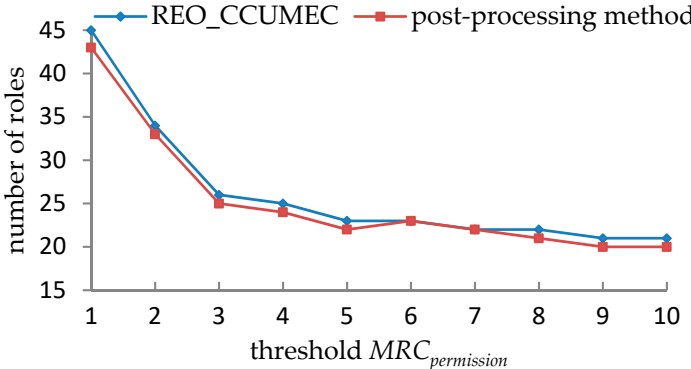

**Figure 5.** Performance comparison under the PCC in Domino.

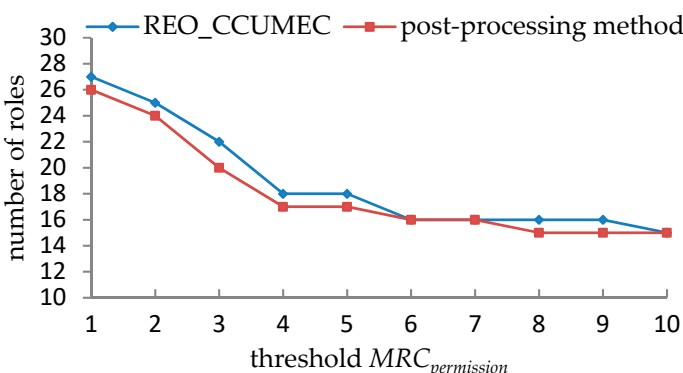

**Figure 6.** Performance comparison under the PCC in Healthcare.

5.2.2. Performance Evaluations under the Double Constraints

Next, we study the impacts of both the UCC and PCC on the number of the optimized roles. According to Algorithm 6, the sets of users or permissions violating the UCC or PCC are first determined. Either a violating user or a violating permission is chosen based on the heuristic strategy in line 4 of the algorithm. In this paper, we consider four heuristics: (1) choose the user or permission with the maximum number based on ($count\_user\_roles(u)$-$MRC_{user}$) or ($count\_perm\_roles(p)$-$MRC_{permission}$), (2) choose the user or permission with the minimum number based on ($count\_user\_roles(u)$-$MRC_{user}$) or ($count\_perm\_roles(p)$-$MRC_{permission}$), (3) choose the violating permission first and then the user, and (4) choose the violating user first and then the permission.

In order to evaluate the effectiveness of the REO_CCUMEC under these double constraints, we implement the experiments with different heuristics on the America-large, Apj, Firewall1, and Firewall2 datasets, as shown in Table 18. The results are shown in Tables 19–22 as the values of

$MRC_{user}$ and $MRC_{permission}$ vary, where the intersections between $MRC_{permission}$ on the row and $MRC_{user}$ on the column represent the number of roles. As the two constraints are mutually exclusive, it is possible that both constraints cannot be satisfied simultaneously, and we use "x" to denote that no valid set of roles is generated.

**Table 19.** The number of optimized roles in the America-large dataset.

| $MRC_{permission}$ | $MRC_{user}$ | | | | | | | | | | | | | | | |
|---|---|---|---|---|---|---|---|---|---|---|---|---|---|---|---|---|
| | 6 | | | | 5 | | | | 4 | | | | 3 | | | |
| | (1) | (2) | (3) | (4) | (1) | (2) | (3) | (4) | (1) | (2) | (3) | (4) | (1) | (2) | (3) | (4) |
| 145 | 423 | 423 | 423 | 423 | 424 | 424 | 424 | 424 | 425 | 425 | 425 | 425 | x | x | x | x |
| 140 | 424 | 424 | 425 | 425 | 425 | 425 | 426 | 426 | 427 | 428 | 428 | 428 | x | x | x | x |
| 130 | 424 | 424 | 425 | 425 | 425 | 425 | 426 | 426 | 427 | 428 | 428 | 428 | x | x | x | x |
| 120 | 425 | 427 | 427 | 427 | 426 | 428 | 428 | 428 | 427 | 431 | 429 | 429 | x | x | x | x |
| 110 | 427 | 431 | 428 | 428 | 428 | 433 | 429 | 429 | 427 | 433 | 431 | 431 | x | x | x | x |
| 100 | 428 | 435 | 431 | 431 | 429 | 437 | 432 | 432 | 433 | 437 | 435 | 435 | x | x | x | x |

**Table 20.** The number of optimized roles in Apj.

| $MRC_{permission}$ | $MRC_{user}$ | | | | | | | | | | | | | | | |
|---|---|---|---|---|---|---|---|---|---|---|---|---|---|---|---|---|
| | 13 | | | | 11 | | | | 9 | | | | 7 | | | |
| | (1) | (2) | (3) | (4) | (1) | (2) | (3) | (4) | (1) | (2) | (3) | (4) | (1) | (2) | (3) | (4) |
| 69 | 456 | 456 | 456 | 456 | 457 | 457 | 457 | 457 | 457 | 459 | 458 | 458 | 461 | 463 | 465 | 465 |
| 65 | 457 | 457 | 457 | 457 | 458 | 458 | 458 | 458 | 458 | 460 | 459 | 459 | 462 | 464 | 466 | 466 |
| 55 | 458 | 460 | 459 | 459 | 459 | 461 | 460 | 460 | 461 | 463 | 461 | 461 | x | x | 467 | 469 |
| 45 | 459 | 462 | 460 | 460 | 460 | 463 | 461 | 461 | 463 | 467 | 462 | 462 | x | x | x | x |
| 35 | 460 | 462 | 460 | 460 | 461 | 463 | 462 | 462 | x | 468 | x | x | x | x | x | x |
| 25 | 460 | 463 | 462 | 462 | 461 | 463 | 462 | 462 | x | 469 | x | x | x | x | x | x |

**Table 21.** The number of optimized roles in Firewall1.

| $MRC_{permission}$ | $MRC_{user}$ | | | | | | | | | | | | | | | |
|---|---|---|---|---|---|---|---|---|---|---|---|---|---|---|---|---|
| | 21 | | | | 17 | | | | 13 | | | | 9 | | | |
| | (1) | (2) | (3) | (4) | (1) | (2) | (3) | (4) | (1) | (2) | (3) | (4) | (1) | (2) | (3) | (4) |
| 27 | 69 | 69 | 69 | 69 | 70 | 70 | 70 | 70 | 71 | 71 | 71 | 71 | 73 | 75 | 73 | 73 |
| 25 | 70 | 70 | 70 | 70 | 71 | 71 | 71 | 71 | 72 | 72 | 72 | 72 | 74 | 76 | 74 | 74 |
| 22 | 70 | 71 | 71 | 71 | 72 | 73 | 72 | 72 | 73 | 73 | 74 | 74 | x | 77 | 75 | 75 |
| 18 | 71 | 72 | 73 | 73 | 73 | 74 | 74 | 74 | 74 | 75 | 76 | 76 | x | x | x | x |
| 15 | 72 | 73 | 74 | 74 | 74 | 75 | 75 | 75 | 75 | 76 | 77 | 77 | x | x | x | x |
| 11 | 73 | 74 | 75 | 75 | 75 | 76 | 76 | 76 | x | 77 | 78 | 78 | x | x | x | x |

**Table 22.** The number of optimized roles in Firewall2.

| $MRC_{permission}$ | $MRC_{user}$ | | | | | | | | | | | | | | | |
|---|---|---|---|---|---|---|---|---|---|---|---|---|---|---|---|---|
| | 9 | | | | 8 | | | | 7 | | | | 6 | | | |
| | (1) | (2) | (3) | (4) | (1) | (2) | (3) | (4) | (1) | (2) | (3) | (4) | (1) | (2) | (3) | (4) |
| 3 | 10 | 10 | 10 | 10 | 11 | 11 | 11 | 11 | 11 | 11 | 11 | 11 | x | x | x | x |
| 2 | x | x | x | x | x | x | x | x | x | x | x | x | x | x | x | x |

It is observed in Tables 19–22 that, when the values of $MRC_{permission}$ are fixed at 145, 69, 27, and 3, respectively, the best experimental results of our method are as follows. As $MRC_{user}$ decreases, 423, 424, and 425 roles are generated from the America-large dataset; 456, 457, 459, and 461 roles are generated from the Apj dataset; 69, 70, 71, and 73 roles are generated from the Firewall dataset; 10, 11, and 11

roles are generated from the Firewall2 dataset. Note that the number of roles increases slightly or remains unchanged as the value of $MRC_{user}$ or $MRC_{permission}$ decreases.

Furthermore, Table 19 shows that the effective sets of roles are generated when $MRC_{user}$ exceeds 3 as $MRC_{permission}$ varies. However, no valid roles are generated from the America-large dataset when $MRC_{user}$ equals 3. Tables 20–22 show that the effective sets of roles are generated when the values of both $MRC_{user}$ and $MRC_{permission}$ are greater, but no valid roles exist with respect to any heuristic strategy when $MRC_{user}$ or $MRC_{permission}$ becomes smaller. In conclusion, for the given $MRC_{user}$ and $MRC_{permission}$, if no valid roles can be generated, the role optimization remains ineffective by reducing one or more constraint values; otherwise, the role optimization remains effective by increasing one or more constraint values.

### 5.3. The Efficiency of the REO_CCUMEC

#### 5.3.1. Experimental Setup

To simulate the actual scenarios while satisfying the security requirements in the role-engineering system, we adopt the method of generating the *t-t* SMER constraints [41]. The value of the cardinality constraint is greater than or equal to 2, and the density of the user-capability matrix changes from 0.4 to 0.6 with a step of 0.05. In addition, the role assigning algorithm is written in Java.

#### 5.3.2. Evaluation Measure

To evaluate the efficiency of the REO_CCUMEC, we compare $RR$ as the cardinality constraints; the *t-t* SMER constraints and the user-capability constraints vary in different datasets.

#### 5.3.3. Experimental Results and Analyses

We use different parameters that include the threshold $MRC_{user}$, the density $\delta$ of matrix $UC$, and set $C$ of *t-t* SMER constraints as inputs, implement the experiments 10 times on the Apj and Customer datasets (as shown in Table 18), and take the median value. The results of the experiments are shown in Figures 7 and 8, where the vertical axis represents the varying values of $RR$, and the lateral axis represents varying values of $MRC_{user}$, $C$, and $\delta$, respectively.

Figure 7 shows that the role-utilization ratio varies with varying values of the cardinality constraint; when the number of set $C$ changes from 100 to 400, the number of roles in the $UC$ constraint is 15, and the density of matrix $UC$ is fixed at 0.6. Specifically, it is observed that the value of the role-utilization ratio first increases with an increase in the value of $MRC_{user}$ and then tends to be stable and no longer changes after a certain point. The reason for this result is that the upper bound of the role-utilization ratio is directly proportional to the value of the cardinality constraint, as shown in Statement 2 of this paper. However, after a particular value for the cardinality constraint, the increase in utilization will be saturated ahead of time, as the *t-t* SMER constraints play important roles in restricting the role assignments. A further observation is that the value of the role-utilization ratio decreases with an increase in the number of the *t-t* SMER constraints because the greater the constraints, the fewer the users assigned to a particular role.

Figure 8 shows that the role-utilization ratio varies with varying values of the density of matrix $UC$; when the number of set $C$ changes from 100 to 400, the number of roles in the $UC$ constraint is 15, and $MRC_{user}$ is fixed at 5. It is observed that the value of the role-utilization ratio decreases with an increase in the value of $\delta$ because the upper bound of the role-utilization ratio is inversely proportional to the value of $\delta$. We also note that this value decreases as the number of the *t-t* SMER constraints increases.

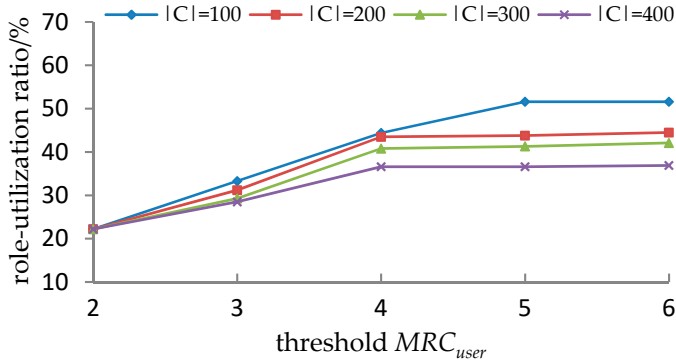

**Figure 7.** Performance of our method with a different $MRC_{user}$ and $C$.

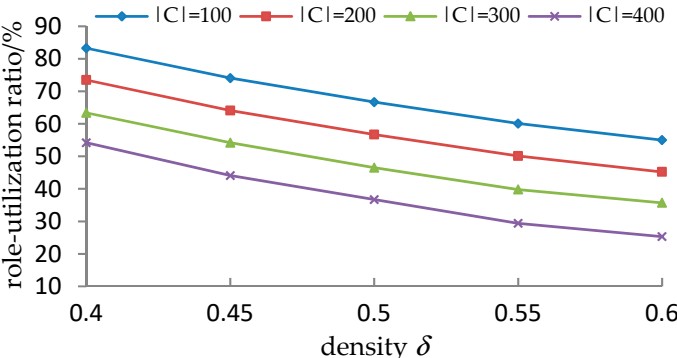

**Figure 8.** Performance of our method with different $\delta$ values.

### 5.4. Advantages and Limitations of the REO_CCUMEC

From the above analyses of the REO_CCUMEC, we find that it has the following main advantages:

(1) In the preprocessing phase, it can reduce the mining scale, while eliminating the redundancies of the mining roles by using partitioning and compressing technologies.

(2) In the role optimization phase, REO_CCUMEC constructs a role-engineering system based on the mining results in the previous phase. Thus, it can satisfy two cardinality constraints simultaneously, and the problem of constraint conflicts between the UCC and PCC can be effectively solved.

(3) In the role assignment phase, besides the cardinality constraints and the given user-capability constraints, we construct *t-t* SMER constraints using the existing mature methods. It is effective and efficient to implement the maximal role assignments, while satisfying all the constraints in the constructed RBAC system.

Meanwhile, it is observed in Sections 1 and 2 that the methods proposed by Kumar et al. [35] and Blundo et al. [36] only satisfied the cardinality constraint RPC; the method proposed by Hingankar et al. [37] only satisfied the RUC; the CPA and RPA proposed by John et al. [38] only satisfied the UCC; the method proposed by Ma et al. [26] satisfied the RUC or RPC; the methods proposed by Sarana et al. [40] did not satisfy any cardinality constraint but satisfied the SMER constraints; the methods proposed by Harika et al. [39] could satisfy the UCC and PCC simultaneously. In addition, the system status was unknown using any of these methods. Although, the method proposed by Roy et al. [28] satisfied the UCC, SMER, and user-capability constraints simultaneously, the RBAC system existed in advance. Therefore, compared with existing studies, the security characteristics of the proposed method are shown in Table 23, where a tick $\sqrt{}$ denotes that the characteristic is available.

**Table 23.** Comparison of security characteristics.

| Characteristic | Kumar et al. [35] Blundo et al. [36] | Hingankar et al. [37] | John et al. [38] | Ma et al. [26] | Sarana et al. [40] | Harika et al. [39] | Roy et al. [28] | Proposed Method |
|---|---|---|---|---|---|---|---|---|
| UCC | | | √ | | | √ | √ | √ |
| PCC | | | | | | √ | | √ |
| RUC | | √ | | √ | | | | |
| RPC | √ | | | √ | | | | |
| SMER | | | | | √ | | √ | √ |
| User-Capability Constraints | | | | | | | √ | √ |
| Unknown System Status | √ | √ | √ | √ | √ | √ | | √ |

Nevertheless, the REO_CCUMEC still has limitations:

(1)  It is observed from Section 5.1.1 that for the given user cluster, how to set the parameters (including compression ratio and the threshold of the support degree) lacks a theoretical justification. Different parameters may cause different evaluation results. Although, the preprocessing roles are very similar to the initial roles from the viewpoint of $sim_P(PA_{compressed}, PA_{initial})$, they are less accurate from the viewpoint of $sim_U(UA_{compressed}, UA_{initial})$ when the threshold $t$ exceeds a particular value.

(2)  It is observed in from Tables 19–22 that the effective roles that can be generated as $MRC_{user}$ and $MRC_{permission}$ vary. However, certain combinations of the values of $MRC_{user}$ and $MRC_{permission}$ cannot produce a valid solution since the UCC and PCC are mutually exclusive, especially when $MRC_{user}$ or $MRC_{permission}$ becomes smaller.

## 6. Conclusions and Future Work

A novel role-engineering method, REO_CCUMEC, has been proposed in this paper. We first converted the basic role mining problem into a clustering problem, and used the partitioning and compressing technologies to eliminate redundancies. We then presented three role-optimization problems with single or double cardinality constraints, and proposed the corresponding algorithms. Lastly, the maximal role-assignments problem was discussed in the constructed role-engineering system. As a result, the proposed method could address the stated problems: Reducing the mining scale and computational complexity in role mining, satisfying the double cardinality constraints simultaneously in the role optimization, and meeting multiple security constraints in the role assignments. The experiments demonstrated that the proposed method is accurate, effective, and efficient.

There are still, however, a few interesting issues to be resolved. In view of the above limitations of the REO_CCUMEC, to further enhance the accuracy of the preprocessing results, one issue is to consider how to provide theoretical justifications for choosing the compression ratio and threshold of the support degree. In order to further enhance the effectiveness of role optimization for satisfying the UCC and PCC simultaneously, another issue is to consider how to determine the upper and lower bounds of constraint values. Moreover, implementing the REO_CCUMEC in systems with the recent hot fields like the blockchain, wireless sensor networks, and internet of things, is also an interesting topic for future work.

**Author Contributions:** Conceptualization, W.S.; methodology, W.S. and H.L.; validation, W.S. and H.L.; formal analysis, W.S.; data curation, H.S.; writing—original draft preparation, W.S.

**Funding:** This work was partially supported by the Natural Science Foundation of China (61501393), the Natural Science Foundation of Henan Province of China (182300410145, 182102210132), and the Key Scientific Research Project of Henan Province University (20B520031).

**Conflicts of Interest:** The authors declare no conflict of interest.

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
