# Peer review of "Role-Engineering Optimization with Cardinality Constraints and User-Oriented Mutually Exclusive Constraints"

_information, doi:10.3390/info10110342_

Round 1

Reviewer 1 Report

The paper is focused on role-engineering optimization with cardinality constraints and user-oriented mutually exclusive constraints. In this paper, according to the Authors, a novel method called role-engineering optimization with cardinality constraints and user-oriented mutually exclusive constraints (REO_CCUMEC) was proposed. The experiments were added to demonstrate the accuracy, effectiveness, and efficiency of the proposed method. The topic is interesting and the paper is well corresponding to the journal aim and scope.

The paper is well structured.

However, there are shortcomings in this paper:

The abstract seems to be too long and too much detailed. The subsection 2.1 is quite short – please consider rebuilding this part and try to combine this content with 2.1.1 and 2.1.2 subsections, based on section 2.2. The section 3.2.2 should start from new page. Justification for the studies considered in Table 23 should be provided. The readers must be sure that they cover "up to date"references The conclusions should describe the finite results - so the used tense should be modified. In section 1 Authors provided three main contributions, however, they did not mention about highlights in conclusions. The limitations are missing. The description of future work requires extension.

The paper is quite well written in English language. Please move single letters to the next line (Line 72,86).

Author Response

Thanks for your helpful comments, and I have revised the paper. Please see the attachment.

Reviewer 2 Report

This work introduces a role-engineering method, called REO_CCUMEC, to reduce the mining scale or computational complexity. They first convert the basic role mining problem into a clustering problem and use the partitioning and compressing technologies to eliminate redundancies. They then proposes several algorithms, i.e., role-assignments algorithm to satisfy multiple constraints.

Good points:

+ Paper structure is okay

+ REO_CCUMEC and several algorithms have proposed

+ A running example is given

Major issues:

- In related work, there is a need to compare the proposed idea with existing ones.

- Writing can be checked.

Author Response

(The authors gave the same response as above.)
